# PET Radiotracers for CNS-Adrenergic Receptors: Developments and Perspectives

**DOI:** 10.3390/molecules25174017

**Published:** 2020-09-03

**Authors:** Santosh Reddy Alluri, Sung Won Kim, Nora D. Volkow, Kun-Eek Kil

**Affiliations:** 1University of Missouri Research Reactor, University of Missouri, Columbia, MO 65211-5110, USA; srad3k@missouri.edu; 2Laboratory of Neuroimaging, National Institute on Alcohol Abuse and Alcoholism, National Institutes of Health, Bethesda, MD 20892-1013, USA; kims8@mail.nih.gov; 3National Institute on Drug Abuse, National Institutes of Health, Bethesda, MD 20892-1013, USA; 4Department of Veterinary Medicine and Surgery, University of Missouri, Columbia, MO 65211, USA

**Keywords:** adrenergic receptor, positron emission tomography, radiotracer

## Abstract

Epinephrine (E) and norepinephrine (NE) play diverse roles in our body’s physiology. In addition to their role in the peripheral nervous system (PNS), E/NE systems including their receptors are critical to the central nervous system (CNS) and to mental health. Various antipsychotics, antidepressants, and psychostimulants exert their influence partially through different subtypes of adrenergic receptors (ARs). Despite the potential of pharmacological applications and long history of research related to E/NE systems, research efforts to identify the roles of ARs in the human brain taking advantage of imaging have been limited by the lack of subtype specific ligands for ARs and brain penetrability issues. This review provides an overview of the development of positron emission tomography (PET) radiotracers for in vivo imaging of AR system in the brain.

## 1. Introduction

Positron emission tomography (PET) is a noninvasive and highly sensitive in vivo imaging technique that uses small amounts of radiotracers to detect the concentration of relevant biomarkers in tissues such as receptors, enzymes, and transporters. These radiotracers can be used to characterize neurochemical changes in neuropsychiatric diseases and also to measure drug pharmacokinetics and pharmacodynamics directly in the human body including the brain [1,2]. PET technology requires positron-emitting radioisotopes, a radiotracer synthesis unit, a PET scanner, and data acquisition components. Among the PET isotopes, cyclotron produced carbon-11 (C-11, *t*_1/2_ = 20.34 min), nitrogen-13 (N-13, *t*_1/2_ = 9.96 min), and fluorine-18 (F-18, *t*_1/2_ = 109.77 min) are frequently used for PET-neuroimaging. PET imaging is particularly valuable to characterize investigational drugs and their target proteins, providing a valuable tool for clinical neuroscience [3].

PET radiotracers for the central nervous system (CNS) should have proper pharmacokinetic profile in the brain and to achieve this property, they were designed to meet five molecular properties: (a) molecular weight <500 kDa, (b) Log D_7.4_ between ~1 to 3 (lipophilicity factor), (c) number of hydrogen bond donors <5, (d) number of hydrogen bond acceptors <10, and (e) topological polar surface area <90 Å^2^. Otherwise, they may either not cross the blood-brain barrier (BBB) or show lack of specific signal due to high nonspecific binding [3,4,5]. In addition, an ideal PET radiotracer should have, though unachievable, (a) high affinity (preferably subnanomolar range) for its target, (b) high selectivity between subtypes or void on off-targets, (c) high dynamic range in specific binding, (d) appropriate metabolic profile, (e) no adverse toxicology effects, and (f) kinetics suitable for mathematical modelling, which requires fast transfer to BBB and clearance in non-target tissues [3]. In general, a radiotracer with binding potential [ratio of target density (*B*_max_) to ligand’s affinity (*K*_d_ or *K*_i_)] value of at least 10 is expected to provide a reliable specific signal in vivo [6,7]. Furthermore, there are some technical challenges associated with CNS PET radiotracer development, which include: limited synthesis time and/or complex syntheses to prepare a radiotracer, attainment of high molar activity, accurate radiometric metabolite analysis. In addition, the radiotracer ought to ideally address targets that are relevant to brain function and for the diagnosis and therapy of a disease [3,8]. Epinephrine (E) and norepinephrine (NE) were discovered in 1894 and 1907, respectively. They are both neurotransmitters and hormones that belong to the group of catecholamines crucial to the function of the body and brain [9,10,11]. Neither E nor NE crosses the BBB but they are synthetized in the brain. NE is synthesized in synapse from *L*-phenylalanine in four steps enzymatically through phenylalanine hydroxylase (PAH), tyrosine hydroxylase, dopa decarboxylase (DDC), and dopamine beta-hydroxylase (DBH), respectively. Further, NE is methylated to convert into E with phenylethanolamine-*N*-methyl transferase (PNMT) [12,13,14]. NE in synaptic cleft is removed either by reuptake via NE transporter (NET) or via metabolism by monoamine oxidase A (MAO-A) or catechol-*O*-methyl transferase (COMT) into various transitional metabolites. Though the noradrenergic neurons are confined to a few relatively small brain areas such as midbrain, pons, locus coeruleus, caudal ventrolateral nucleus, and medulla, they send extensive projections to most brain regions [15]. Table 1 summarizes the brain distribution (reported in mice) and function of various AR subtypes and their involvement in some brain disorders for CNS-ARs.

The effects of NE and E in the CNS and PNS is mediated mainly through two main classes of adrenergic receptors (ARs): alpha-ARs (α-ARs) and beta-ARs (β-ARs) [16]. The ARs were first identified in 1948 and pharmacological and molecular cloning techniques since then have identified various subclasses of ARs [17,18]. α-ARs are divided into α1 and α2 subclasses, wherein, each of these has three subtypes: α1A, α1B, α1D and α2A, α2B, α2C. The three subtypes of β-ARs are β1, β2, and β3 [19]. These receptors in the CNS are G-protein coupled receptors (GPCR) and are implicated in pathophysiology of various diseases, biochemical pathways, and biological functions [18].

**Table 1 molecules-25-04017-t001:** Brain distribution of AR subtypes and their associated brain disorders.

Receptor		Distribution	Distinct Functions and Associated Disorders	Ref
α1	α1A	High levels in olfactory system, hypothalamic nuclei, and brainstem. Moderate levels in amygdala, cerebral cortex, and cerebellum	Involved in neurotransmission of NE as well as γ-aminobutyric acid (GABA) and NMDA. May mediate effects of anti-depressants in treating depression and obsessive compulsive disorder (OCD)	[18,20,21,22,23,24]
	α1B	Thalamic nuclei, lateral nucleus of amygdala, cerebral cortex, some septal regions, brain stem regions	May play a role in behavioral activation. Associated with addiction, and neurodegenerative disorders (Multiple System Atrophy)	[18,20,21,24,25,26,27]
	α1D	Olfactory bulb, cerebral cortex, hippocampus, reticular thalamic nuclei, and amygdala	Mediates changes in locomotor behaviors. Associated with stress.	[18,20,23,28,29]
α2	α2A	Locus coeruleus, midbrain, hypothalamus, amygdala, cerebral cortex, and brain stem	Mediate functions of most of the α2-agonists used in sedation, antinociception, and behavioral actions. Associated with ADHD, anxiety	[18,23,30,31,32,33,34,35]
	α2B	Thalamus, hypothalamus, cerebellar Purkinje layer	Mediate antinociceptive action of nitrous oxide	[18,30,31]
	α2C	Hippocampus, striatum, olfactory tubercle, medulla, and basal ganglia	Involved in the neuronal release of NE as well as dopamine and serotonin. Potential therapeutic targets in depression & schizophrenia	[18,30,31,36,37,38,39]
β	β1	Homologous distribution. Expression was found (mostly β1 and β2) in frontal cortex, striatum, thalamus, putamen, amygdala, cerebellum, cerebral cortex and hippocampus.	Essential to motor learning, emotional memory storage and regulation of neuronal regeneration. Associated with mood disorders, aging, Alzheimer’s disease, Parkinson’s disease.	[16,18,40,41,42,43,44,45,46,47,48]
	β2			
	β3			

Given its biological significance, the adrenergic system has emerged as an important target for PET studies. Although ARs play major roles in the brain, most PET studies of ARs have focused on cardiac imaging [49,50] and PET studies of CNS-ARs are very limited. The other components of adrenergic system such as NET and MAO-A were studied using PET. Various C-11 and F-18 radiotracers of NET-selective anti-depressants (e.g., reboxetine) were examined using PET to monitor the function of noradrenergic system in CNS. Radiotracers, for example, (*S,S*)-[^11^C]*O*-methyl-reboxetine ([^11^C]MRB) and [^18^F]FMeNER-D_2_, were shown to exhibit desirable in vivo properties and their regional distribution in the brain is consistent with known distribution of NET in preclinical/clinical settings [51,52,53,54]. These radiotracers have been employed to monitor NET availability in different related diseases, including obesity, major depressive disorder, and Parkinson’s disease. Likewise, radiotracers, for instance, [^11^C]Harmine demonstrated clinical success for in vivo brain imaging of MAO-A, respectively [55,56,57]. [^11^C]Harmine has been applied in several-PET neuroimaging studies to study the role of MAO-A in different pathological conditions, including nicotine/alcohol dependence, and Alzheimer’s disease.

However, the metabolic functions of MAO-A do not necessarily reflect the activities of adrenergic system as they also metabolize other neurotransmitters, such as, dopamine and serotonin (5-HT). Therefore, the in vivo activities of adrenergic neurons, thus far, were exclusively examined by PET imaging based on NET. Since the activities of NET reflect only on presynaptic systems of adrenergic neurons, novel PET radiotracers that can observe the features of postsynaptic adrenergic system are still required. PET-neuroimaging of AR subtypes combined with NET can scrutinize the unique implications of adrenergic system in various pathophysiological conditions of the brain. Subtype specific PET radiotracers for CNS-ARs have the potential to help clarify the roles of ARs in brain pathophysiology and provide suggestions towards the diagnosis and treatment for diseases such as depression, attention deficit hyperactivity disorder (ADHD), substance use disorders, schizophrenia, and neurodegenerative diseases that involve ARs. This review, based on reports published through 2019, summarizes the development of PET radioligands for CNS-ARs in animal models and human subjects and presents suggestions for further development for CNS-AR radiotracers.

## 2. α1-AR PET Radiotracers

Three highly homologous subunits of α-1 ARs, α1A, α1B, and α1D, have been shown to have different amino acid sequence, pharmacological properties, and tissue distributions [58,59,60]. A detailed review of α1-AR pharmacology is given by Michael and Perez [58]. Binding of NE/E to any of the α1-AR subtypes is stimulatory and activates G_q/11_-signalling pathway, which involves phospholipase C activation, generation of secondary messengers, inositol triphosphate and diacyl glycerol and intracellular calcium mobilization. In PNS, as well as in the brain’s vascular system it results in smooth muscle contraction and vasoconstriction. While the signaling effects of α1-ARs in the cardiovascular system are well studied [61,62], the role of α1-ARs in CNS is complex and not completely understood. The Bmax values of α-1 ARs were measured from saturation assays using [^3^H]prazosin (a selective α-1 blocker) with tissue homogenates from rats and the observed binding capacities (fmol/mg tissue) of prazosin in cortex, hippocampus, and cerebellum were 14.49 ± 0.38, 11.03 ± 0.39, and 7.72 ± 0.11, respectively [23,63]. The α1-ARs are postsynaptic receptors and can also modulate release of NE. In the human brain, α1-AR subtypes are localized in amygdala, cerebellum, thalamus, hippocampus, and to some extent in striatum [20,22,64].

The anti-depressant effects of noradrenergic enhancing drugs as well as their effects on anxiety and stress reactivity points to their relevance of α1A, α1D to these behaviors [21,28,65]. Furthermore, α1A-ARs regulate GABAergic and NMDAergic neurons [20]. A decrease in α1A-AR mRNA expression was observed in prefrontal cortex of subjects with dementia [66,67]. The α1B-ARs has also been shown to be crucial to brain function and disease [18]. For example, α1B-knockout (KO) mice study revealed that α1B-AR modulates behavior, showing increased reaction to novel situations [24,27]. In addition, the locomotor and rewarding effects of psychostimulants and opiates were decreased in α1B-KO mice, highlighting their role in the pharmacological effects of these drugs [24]. On the other hand, studies using the α1B-overexpression model suggest their involvement in neurodegenerative diseases [25].

Several α1-AR agonists and antagonists (α1-blockers) are available in the market as drugs to treat various heart and brain disorders [68,69]. Pharmacology of most of these drugs is complicated by the fact that they have strong affinities for other receptor systems, such as serotonin and dopamine receptors. The need to develop α1-AR selective/α1-AR subtype specific drugs is demanding. Undoubtedly, PET radiotracers selective for α1-AR are valuable to assess α1-ARs contribution to brain function and disease. In the late 1980s, prazosin, used for the treatment of hypertension, was labelled with carbon-11 to image α1-ARs with PET [49]. Following this, [^11^C]prazosin analogous, [^11^C]bunazosin and [^11^C]GB67 (Figure 1A) were developed as PET radiotracers to image α1-ARs in the cardiovascular system [49,70]. These tracers were shown to have limited BBB permeability and were deemed to be not suitable for PET-neuroimaging. Efforts to develop PET radiotracers to image α1-ARs in the CNS was mainly based on antipsychotic drugs such as clozapine, sertindole, olanzapine, and risperidone that have mixed binding affinities for D_2_, 5HT_2A_ receptors and α1-ARs in the nanomolar range. Their affinity for α1-ARs has been shown to contribute to antipsychotic efficacy uncovering their role in psychoses [71,72].

Two C-11 labelled analogs of the atypical antipsychotic drug sertindole were first reported by Balle et al. in the early 2000s (Figure 1B) [73,74]. Several other analogs with good affinities (*K*_i_ < 10 nM) for α1-ARs were subsequently reported by the same group. Two analogs, in which chlorine in sertindole (**1**) is replaced by a 2-methyl-tetrazol-5-yl (**2**) and a 1-methyl-1,2,3-triazol-4-yl (**3**), were labelled with C-11 [73,74]. The in vitro affinities (*K*_i_) of **1**, **2** and **3** for α1-ARs are 1.4, 1.8 and 9.5 nM, respectively. Both [^11^C]**2** and [^11^C]**3** were prepared using ^11^C-methylation with [^11^C]methyl triflate from their corresponding *N*-desmethyl precursors. The molar activity of [^11^C]**2** and [^11^C]**3** was reported at 70 GBq/μmol and 15 GBq/μmol, respectively. Their brain distribution examined with PET in Cynomolgus monkeys showed that brain uptake of [^11^C]**2** and [^11^C]**3** was slow and low, with [^11^C]**3** showing somewhat higher brain uptake than [^11^C]**2**. Their brain distribution was homogenous and specific binding to α1-ARs could not be demonstrated. It was also concluded that these two radiotracers were not suitable to image α1-ARs in brain owing to rapid metabolism, substantial distribution to other organs, and substrates for active efflux mechanism.

In the early 2010s, further optimization of structure-activity relationship (SAR) studies led to the identification of two more sertindole analogs, **4** and **5** (Lu AA27122), with higher α1-AR selectivity over D_2_ receptors (Figure 1B). Compared to the in vitro affinity for α1A-AR subtype (*K*_i_ = 0.16, 0.52 nM for **4** and **5**), the affinity for the α1B and α1D subtypes is 4 to 15 times less potent (Table 2). The LogD_7.4_ values for **4** (2.7) and **5** (1.9) were in the optimal range for in vivo brain imaging [75,76,77]. Both [^11^C]**4** and [^11^C]**5** were prepared in a similar manner like above with >370 GBq/μmol molar activity for non-human primate studies [75]. Interestingly, while [^11^C]**4** showed very poor brain uptake, [^11^C]**5** had suitable brain uptake (4.6% ID/cc at 36 min). However, its binding was not blocked with a pharmacological dose of prazosin pretreatment, indicating lack of α1-AR specificity.

At the same time, another C-11 labelled radiotracer, [^11^C]**7** ([^11^C]Lu AE43936, Figure 1C) for brain α1-ARs was developed and evaluated by Risgaard et al. [76]. This radiotracer was based on the antipsychotic octoclothepin (**6**, Figure 1C), which belongs to the tricylic dibenzotheiepin group and has inverse agonist effects at dopamine, serotonin, and α1 receptor sites [78]. Two enantiomers of **7** (*R*/*S*) were radiolabelled on the basis of their varying selectivity and specificity for α1-AR subtypes (Table 2) and imaged with PET in female Danish Landrace pigs. The baseline PET imaging results indicated that neither of the radiolabelled isomers entered the pig’s brain. Pretreatment with cyclosporin A (CsA) [79] increased the brain uptake of (*R*)-**7** in α1-AR rich cortex, thalamus (above 2 SUV), suggesting that (*R*)-**7** was a substrate for active efflux transporters. Further cell studies specified that (*R*)-**7** is a substrate for p-glycoprotein (Pgp).

So far, none of the reported radiotracers showed promising results for in vivo brain imaging of α1-ARs for they were limited by poor BBB penetration, being substrates of Pgp or lack of binding specificity. Therefore, novel α1-AR radiotracers that overcome brain permeability and display good affinity and subtype-selectivity are required to evaluate the role of α1-AR subtypes in brain pathophysiology. Several α1-AR subtype specific (high affinity for one subtype over the other) and nonspecific compounds have been reported over the years [26,29,80,81]. Either direct radiolabeling (if possible) or chemical modification and then radiolabeling of the most specific compounds could be an optimal approach for developing new α1-subtype specific radiotracer, considering the aforementioned CNS-PET radiotracer criteria.

## 3. α2-AR Subtype and Nonspecific PET Radiotracers

Unlike α1-ARs, α2-ARs decrease adenyl cyclase activity in association with G_i_ heterotrimeric G-protein and hence are inhibitory. They mediate many NE effects including cognition and readiness for action [11,30]. The brain distribution of the three subtypes, α2A, α2B, and α2C, was characterized by autoradiography and immunohistochemistry techniques. Among the three subtypes, α2A-ARs are the most abundant in the brain and localized in locus coeruleus, midbrain, hippocampus, hypothalamus, amygdala, cerebral cortex and brain stem. The α2B-ARs are located in thalamus and hypothalamus and α2C-ARs in cortex, hippocampus, olfactory tubercle and basal ganglia [51,52,53]. In the mouse brain, ~90% of α2-ARs are α2A-ARs and ~10% are α2C-ARs [30,31,32,82]. Notably, binding experiments using [^3^H]2-methoxyidazoxan (a selective α-2 AR antagonist) with postmortem human brain detected 100% of α2A-ARs population in the hippocampus, cerebellum, and brainstem (*B*_max_ = 34–90 fmol/mg protein). In addition to this, α2A-AR (*B*_max_ = 53 fmol/mg of protein) and α2B/C-AR (*B*_max_ = 8 fmol/mg of protein) were detected in the frontal cortex [30]. An extensive array of agonists and antagonists for α2-ARs have been developed. The main limitation of these ligands is lack of subtype selectivity for α2-ARs and off-target binding to other receptors [83,84]. α2A-ARs are mostly presynaptic and agonists inhibit NE release from the terminals and are used to treat hypertension, drug withdrawal, and ADHD whereas antagonists increase NE release and are used as antidepressants. Development of α2-AR subtype selective PET tracers would facilitate medication development and help gain further understanding of their role in brain diseases.

During the 1980s, two research groups identified two potent α2-AR selective antagonists via radioligand binding assays. One WY-26703 belongs to the benzoquinolizine class, and the other MK-492 belongs to the benzo[b]furo-quinolizine class [85,86]. Based on these templates, Bylund’s group developed two PET tracers: [^11^C]WY-26703 (**8**) in 1992 and [^11^C]MK-912 (**9**) in 1998 (Figure 2) [87,88]. Both radiotracers were prepared from their respective *N*-desmethyl precursors via ^11^C-methylation with 30.71–34.41 GBq/μmol molar activity. The in vitro binding assays and ex vivo biodistribution studies (tissue dissection followed by γ-counting) in rodents indicated that both radiotracers crossed the BBB and **9** showed higher affinity and specific binding to α2-ARs than **8**. However, PET studies of **8** and **9** in Rhesus monkeys showed fast washout from brain and nonspecific binding; thus it was concluded that they were not appropriate for PET imaging of α2-ARs in brain.

In 1997, Pike’s group developed [^11^C]RS-15385-197 (**10**) and [^11^C]79948-197 (**11**) as PET α2-ARs ligands (Figure 2) [89]. These radiotracers were prepared from their respective *O*-desmethyl precursors through the ^11^C-methylation method with 61 ± 17 GBq/μmol (**10**) and 64 ± 3 GBq/μmol molar activity (**11**). Biodistribution, brain uptake, and metabolic profile studies were done in male Sprague-Dawley rats. They observed specific signals in brain (mainly cerebellum) at 30–90 min (70–95% radioactivity of parent radioligand), which was analogous to their results with [^3^H]**10**. Nonspecific binding in brain was **11** > **10**, which mostly likely reflected their differential metabolism (**11** > **10**). Thus, they chose **10** to quantify α2-ARs in the human brain using PET [90]. Studies in two volunteers with **10** revealed a low brain uptake index (BUI) due to high affinity to human plasma proteins. Consequently, **10** was not studied further.

In 2002, Crouzel’s group chose atipamezole, an α2-AR selective antagonist, to develop [^11^C]atipamezole, **12** (Figure 2) [91]. The radiotracer was prepared through an unique approach using 2-ethyl-2-oxoacetylindane, [^11^C]formaldehyde ([^11^C]HCHO) in the presence of zinc oxide and ammonium hydroxide (similar to Debus-Radziszewski imidazole synthesis) with an overall yield of 1.5%. However, no PET studies were reported with **12**.

Furthermore, in 2002, Smith’s group developed two tetracyclic based anti-depressant radiotracers, mianserin (**13**) and mitrazepine (**14**), that have potent antagonist properties at α2-ARs and also at serotonin receptors (5HT_2A_, 5HT_2C_) and labelled them with C-11 to prepare **13** and **14** (Figure 2) [92,93]. The radiotracers **13** and **14** were prepared from their respective *N*-desmethyl precursors via ^11^C-methylation with ~40 GBq/μmol and 5–7 GBq/μmol molar activity, respectively. PET studies in female pigs with **13** showed limited binding potential in brain whereas **14** showed more favorable properties including slow metabolism, fast brain uptake and sufficient target-to-background ratio for pharmacokinetic parameters estimation.

Radiotracer **14** had higher binding in the frontal cortex, thalamus, and basal ganglia where pretreatment with unlabeled mitrazepine revealed that its binding was reversible, whereas, in the cerebellum and olfactory tubercle, it was not. Notably, using the α2-AR subtype KO mouse model they validated the receptor selectivity of **14**. In 2004 and 2009, this group conducted a clinical trial with volunteers using **14** to study its distribution, metabolism and pharmacokinetics [94,95]. The results revealed that **14** can serve as a PET radiotracer to image α2-ARs in the brain, though identification of its metabolites and its nonselective binding are limitations.

Two years later, Leysen’s group developed a reversible, potent and selective α2-AR antagonist, viz. R107474 (**15**, Figure 3) [96]. They prepared [^11^C]**15** through Pictet-Spengler condensation method using [^11^C]HCHO and the respective secondary amine with 24–28 GBq/μmol molar activity at the end of bombardment (EOB). They carried out ex vivo autoradiography to measure in vivo α2A-ARs and α2C-ARs occupancy of **15** in rats. Biodistribution studies showed rapid uptake of **15** into brain and other tissues with the brain showing the highest uptake other than liver and kidneys. In the brain the highest uptake of **15** was in the septum (3.54 ± 0.52 ID/g) and entorhinal cortex (1.57 ± 0.50 ID/g) whereas the lowest was in the cerebellum, a region with very low density of α2-ARs. However, the potential of **15** was not investigated further.

In 2006, Jacobsen’s group developed C-11 labelled yohimbine (**16**, Figure 3) [97], an antihypertensive agent. Yohimbine has potent antagonist properties at α2-ARs, but also interacts with α1 and 5-hydroxy tryptamine 1A receptors (5-HT_1A_). The radiotracer **16** was prepared through ^11^C-methylation of yohimbinic acid using C-11 methyl iodide ([^11^C]CH_3_I) and obtained with 40 GBq/μmol molar activity.

PET studies were performed in pigs to obtain whole-body and dosimetry recordings and for dynamic brain imaging. Interestingly, no radioactive metabolites of **16** were reported in pig plasma and binding of **16** was observed in α2-AR-rich regions where it was displaceable by co-injection of pharmacological doses of yohimbine or selective α2-AR antagonist (Figure 4). Later, **16** was used to image α2-ARs in the human brain (*n* = 6) using PET [98]. Highest binding of **16** was observed in cortex and hippocampus and the lowest in corpus callosum, which was used as a reference region to estimate the average total distribution (*V*_T_) in other brain regions. The radiotracer **16** seems to be a suitable radiotracer to image α2-ARs but has similar issues as of **14,** which need to be addressed.

The concentration of α2-subtype receptors in the brain is low (5–90 fmol/mg range) increasing the challenge for their detection by PET [31]. Therefore, α2- subtype specific PET tracers (with subnanomolar affinities and >30-fold selectivity) still need to be developed for the quantification of α2-subtype receptors and to assess their role in brain diseases. A few research groups have developed α2-subtype specific PET radiotracers, but success has been limited.

### 3.1. α2A-Specific PET Radiotracers

Kumar’s group in 2010 developed [^11^C]MPTQ (**17**, Figure 5) for the quantification of α2A-ARs in vivo [35]. Compound **17** was shown to have blocking effects on α2A-ARs in vivo in brain and has stronger affinities for α2A-AR (*K*_i_ = 1.6 nM) than α2C-AR (*K*_i_ = 4.5 nM) and 5-HTT (serotonin transporter, *K*_i_ = 16 nM) [99]. They anticipated no binding of **17** to α2B and α2C-ARs since the densities of these receptors are lower than α2A-ARs. In addition, the 10-fold higher affinity of **17** α2A-ARs over 5-HTT is advantageous for α2A-ARs as both have similar *B*_max_ values. The radiosynthesis of **17** was accomplished through ^11^C-methylation of its respective *N*-desmethyl precursor with 74–88.8 GBq/μmol molar activity at the end of synthesis (EOS). PET studies in baboons with **17** showed that it penetrated the BBB and accumulated in α2A-AR-rich brain areas. They ruled out binding of **17** to 5-HTT due to its low uptake in the hippocampus, temporal cortex, and occipital cortex, which are the brain regions with the highest binding of 5-HTT radiotracers. No further studies were reported using **17**.

In search of a selective agonist to α2-ARs, Lehmann’s group identified 1-[(imidazolidin-2-yl)imino]indazole (marsanidine) [100] and later developed an α2A subtype specific ligand by introducing fluorine to marsanidine [33]. The reported binding affinity (*K*_d_) of 6-fluoromarsanidine for α2A (33 nM) is higher than for α2B (72 nM) and α2C (600 nM). Solin et al., in 2019, prepared 6-[^18^F]fluoromarsanidine (**18**, Figure 5) through electrophilic ^18^F-radiofluorination using [^18^F]selectfluor bis(triflate) and a corresponding precursor with 3–26 GBq/μmol molar activity at the EOS [34]. In vivo PET was performed in rats and α2A-KO mice, but the radiotracer was not continued further because of its rapid metabolism and high nonspecific uptake in rat and mouse brain.

### 3.2. α2C-Specific PET Radiotracers

Animal models, such as the forced swimming test (FST) and the prepulse inhibition (PPI) are used to screen for anti-depressants and anti-psychotics, respectively. The use of α2C-KO and α2C-overexpression (α2C-OE) mouse models in FST and PPI paradigms suggested that α2C-specific compounds may have therapeutic benefits for depression and schizophrenia [32,38]. In 2007, Orion pharma from Finland identified an acridine-based compound, JP-1302 [38], and a research group from Japan identified a methyl benzofuran based compound, MBF [101], as selective α2C antagonists. Both these ligands have high affinities for α2C (JP-1302 *K*_i_ = 28 nM, MBF *K_i_* = 20 nM) than for α2A (JP-1302 *K*_i_ = 3500 nM, MBF *K*_i_ = 17,000 nM) and α2B (JP-1302 *K*_i_ = 1500 nM, MBF *K*_i_ = 750 nM). Based on these findings, Zhang’s group, in 2010, synthesized [^11^C]JP-1302 (**19**) and [^11^C]MBF (**20**)(Figure 6) as PET probes to evaluate their BBB penetration and α2C selective binding in the brain [37]. The radiotracers **19** (molar activity 95 ± 24 GBq/μmol) and **20** (molar activity 62 ± 15 GBq/μmol) were prepared using ^11^C-methylation from *N*-desmethyl and *O*-desmethyl precursors, respectively. PET studies were conducted in WT and Pgp, breast cancer resistance protein (BCRP) KO mice using both radiotracers.

This combined KO model is useful to evaluate whether brain penetration of PET probes is sensitive to Pgp and BCRP. After injection of the radiotracers, their levels in the brain were low in WT mice whereas they were higher in Pgp and BCRP KO mice. The regional binding of these radiotracers did not correspond with the regional brain distribution of α2C, so it was concluded that they were inadequate to evaluate α2C-ARs in brain with PET.

In 2014, researchers from Turku PET center and Orion Pharma reported the radiosynthesis of [^11^C]ORM13070 (**21**, Figure 6) with molar activity 690 ± 340 GBq/μmol and its evaluation in rats and in α2A and α2AC KO mice with PET [36,102]. The binding affinities of **21** for α2C (3.8 nM) is higher than for α2A (109 nM) and α2B (23 nM).

The in vivo PET and ex vivo autoradiography of **21** in rat indicated that its brain distribution corresponds to the regional distribution of α2C in brain, with highest levels in striatum and olfactory tubercle. Pretreatment with atipamezole, a α2-sutype nonselective antagonist blocked the binding of **21** into these regions. Furthermore, by using α2A and α2AC KO model mice, they demonstrated α2C specificity of **21**. The brain uptake of **21** in α2A-KO and WT mice was similar whereas, negligible uptake occurred in α2AC KO (Figure 7, left). They represented time-activity curves for striatum and cerebellar cortex of three mice types (Figure 7, right) and the radioactivity ratios at 5–15 min for α2A, α2AC KO mice, and WT mice were 1.51–1.51, 1.06–1.09 and 1.51–1.57, respectively.

Accordingly, **21** was studied in healthy men to estimate its metabolism, pharmacokinetics, whole-body distribution and radiation dosimetry [39]. Good results were obtained in rodent and human PET studies with **21**, except for its fast washout from brain. Better pharmacokinetics, higher affinity, and specificity can potentially be enhanced by structural modifications to **21**. Given that the α2A-ARs are widely distributed in brain in contrast to α2C-ARs, a candidate with subnanomolar affinity for α2C-ARs (>50-fold affinity than α2A-ARs) is needed for a PET radiotracer. As α2C-ARs are of interest as therapeutic targets in brain diseases, the α2C-specifc PET radiotracers would facilitate their development as medications and help in investigations of α2C-ARs in the human brain.

## 4. β-ARs and Nonselective PET Radiotracers

β-ARs are associated with G_s_-heterotrimeric G-protein and mediate intracellular signaling through adenyl cyclase activation and cyclic adenosine monophosphate (cAMP) production. β-ARs are classified into β1, β2, and β3 subtypes, in which, the former two have been much more explored [42,46]. Quite a lot of selective and nonselective β-AR agonists and antagonists (blockers) are available as drugs in the market to treat various cardiac and pulmonary disorders. In the brain, β-ARs are localized in the frontal cortex, striatum, thalamus, putamen, amygdala, cerebellum and hippocampus [48]. The density of β-ARs in brain is sensitive to brain pathophysiology. Notably, the density of β-ARs decrease with age [40]. Light microscopic autoradiography using [^3^H]dihydroaloprenolol (a selective β-blocker) with rat brain sections has shown a wide distribution of β-ARs in forebrain and cerebellum regions (*B*_max_ = 23 fmol/mg tissue) [103]. Similarly, *B*_max_ value of 18 fmol/mg protein was reported in pre-frontal cortex of subjects with Parkinson’s disease [48,104]. By altering the Ca^2+^ levels through *N-*methyl-*D-*aspartate (NMDA) receptors in hippocampus, β-ARs modulate synaptic plasticity, including that needed for memory [44,45]. The blockade of β-ARs is associated with a small increased risk for Alzheimer’s and Parkinson’s disease [43,105]. In addition, abnormal function and densities of β-ARs have been reported in mood disorders and schizophrenia [41,47,106].

Several radioligands, mostly based on β-blockers, were validated for imaging of β-ARs in the heart [50]. The majority of β-blockers possess a hydroxyl propylamine moiety in their structures that is vital for binding to β-ARs and this moiety was maintained in most of these radioligands. PET radiotracers have succeeded in imaging and quantifying myocardial and pulmonary β-ARs in human [107,108], whereas, PET radiotracers for cerebral β-ARs have been more challenging. The clinical PET radiotracers for cardiac β-ARs have negative Log P values (<−2), which is not suitable for imaging the brain. Several lipophilic and high to moderate affinity β-AR nonselective antagonists were explored as PET radiotracers to image β-ARs in the brain.

During the 1980s, propranolol, a β-blocker drug was labelled with C-11 (**22**, Figure 8) but was unsuitable as a PET ligand for β-ARs because of high nonspecific binding in vivo [109,110]. Subsequently, Berridge’s group described the synthesis of two isomers (*R*/*S*) of [^18^F]fluorocarazolol (**23**, Figure 8) through reductive amination using [^18^F]fluoroacetone and desisopropylcarazolol with 18.5–37 GBq/μmol molar activity [111,112]. The radiotracer **23** has subnanomolar *K_i_* values for β-ARs (β1 0.4 nM, β2 0.1 nM) and Log P_7.4_ value of 2.19. The same group used *S*-**23** for PET imaging of the pig heart and lungs to validate the β-AR biding. In 1997, Waarde et al., employed *S*-**23** to image β-ARs in the human brain and obtained positive results [113]. They observed specific binding (blocked with pindolol) of *S*-**23** in β-AR rich areas, striatum and various cortical areas. However, the radiotracer was discontinued for further human studies as fluorocarazolol was positive for the Ames test i.e., mutagenic [114].

Two research groups conducted biodistribution studies in rats using [^18^F]fluoropropranolol (**24**) and [^11^C]ICI 118,551 (**25**) (Figure 8), which failed because of their nonspecific binding [115,116]. In 2001, Fazio’s group described two isomers (*R*/*S*) of C-11 labelled bisprolol (β1 *K*_i_ 1.6 nM, β2 *K*_i_ 100 nM and Log P_7.4_ = −0.2) (**26**, Figure 9) to image β1-ARs in the brain [117]. The radiosynthesis of **26** was accomplished via reductive amination using [^11^C]acetone and desisopropyl bisprolol precursor with 129.5 ± 37 GBq/μmol molar activity at the EOS. They observed little specific uptake of **26** in β1-AR rich regions in the rat’s brain and also high nonspecific uptake in the pituitary (1.8 ± 0.3 ID at 30 min), a region with high β2-ARs levels. No further studies were reported using **26** to image β-ARs.

In 2002, Elsinga’s group reported five different potent and lipophilic β-AR antagonists (carvedilol, pindolol, toliprolol, bupranolol, and penbutolol) as PET probes to image β-ARs in rat brain [118]. C-11 labelled carvedilol (**27**, Figure 9; molar activity 12.97–25.9 GBq/μmol) was prepared through ^11^C-methylation using [^11^C]CH_3_I and its respective *O-*desmethyl precursor, whereas, [^11^C]pindolol (**28**, molar activity 25.9–37 GBq/μmol) and [^11^C]toliprolol (**29**, molar activity 22.2–25.9 GBq/μmol) were prepared via reductive amination using [^11^C]acetone and the respective desisopropyl precursors. The F-18 tracers of bupranolol (**30**, molar activity 11.1–18.5 GBq/μmol) and penbutolol (**31**, 22.2–99.9 GBq/μmol) were also prepared by means of reductive amination but using [^18^F]fluoroacetone and the respective des-fluoro-isopropyl precursors.

The five radiotracers had strong affinities (subnanomolar *K*_d_) for β1 and β2-ARs. Although these radiotracers had sufficient affinity and lipophilicity for in vivo imaging, none showed good brain uptake. This group also evaluated S-[^18^F]fluoroethylcarazolol (**32**, β1 *K*_i_ = 0.5 nM, β2 *K*_i_ = 0.4 nM and Log P_7.4_ = 1.94) for in vivo imaging of β-ARs in rat brain [119]. The radiotracer **32** (Figure 9) was prepared via an epoxide ring-opening using [^18^F]fluoroethylamine and the corresponding epoxide with >10 GBq/μmol molar activity. The radiotracer accumulated in brain with uptake reflecting cerebral β-ARs binding. However, no further PET imaging studies were reported using **32** probably because of its analogous nature to **23** which was shown to be positive Ames test [114,120].

In 2008, Elsinga’s and Vasdev’s groups chose exaprolol (β-AR *K*_d_ = 9–9.5 nM) and developed *S*-[^11^C]exaprolol (**33**) and *S*-[^18^F]fluoroexaprolol (**34**), respectively, to image β-ARs with PET (Figure 10) [120,121]. Radiotracer **33** was prepared via reductive amination using [^11^C]acetone and desisopropylexaprolol precursor with >10 GBq/μmol molar activity and the radiotracer **34** was prepared through a nucleophilic substitution reaction using [^18^F]fluoride and a corresponding tosylate precursor followed by reductive hydrolysis, with 34.29 GBq/μmol molar activity. Regardless of good binding and kinetic properties, both these radiotracers showed high nonspecific uptake in the brain and were found to be inadequate for PET imaging of β-ARs.

Again in 2014, Elsinga’s group developed [^18^F]FPTC (**35**, Figure 10) for PET imaging of β-ARs in brain [122]. The radiotracer **35** is a derivative of carazolol, wherein, isopropylamine group of carazolol was replaced by a PEGylated triazole group. It was prepared through Huisgen’s 1,3-dipolar cycloaddition (click reaction) using F-18 labelled PEGylated alkyne and the corresponding azide with >120 GBq/μmol molar activity. Although **35** was shown to have appropriate LogP_7.4_ (2.48) and specific binding in in vitro assays, it could not visualize β-ARs in the brain, lung or heart using micro-PET.

Thus, the development of PET radiotracers for neuroimaging of β-ARs remains a challenge and as of now, there are no β-AR subtype specific PET radiotracers. Such radiotracers are important to expand our understanding of the role of β-ARs in aging and memory formation and also to assess their function in behavioral disorders. Future research, as suggested by Elsinga and Waarde [48], should consider modifying the imaging agents used for myocardial β-ARs rather than radiolabeling existing β-blocker drugs. Alterations should optimize Log P_7.4_ (2-3), high affinity and selectivity to β-ARs and no substrate affinity for Pgp.

## 5. Conclusions

Over the past four decades, significant efforts have been made to develop CNS-ARs PET ligands for brain imaging. Despite these efforts, very few PET radiotracers are available to selectively image AR subtypes in the brain. The development of specific radiotracers is hindered mainly by the low receptor densities of each AR subclass within the brain, which requires further optimization processes for highly potent and BBB permeable ligands. Though challenging, AR subtype specific agonist/antagonist PET radiotracers are needed to ascertain AR’s role in brain pathophysiology and for medication development.

## Figures and Tables

**Figure 1 molecules-25-04017-f001:**
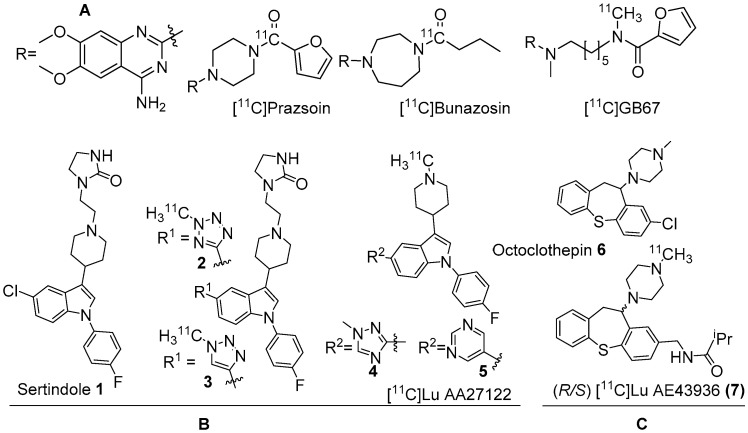
(**A**) Earlier PET radiotracers, [^11^C]Prazosin, [^11^C]Bunazosin, and [^11^C]GB67 for cardiac α1-AR imaging. (**B**) Antagonist PET radiotracers based on sertindole. (**C**) Antagonist PET radiotracers based on octoclothepin for brain α1-AR imaging.

**Figure 2 molecules-25-04017-f002:**
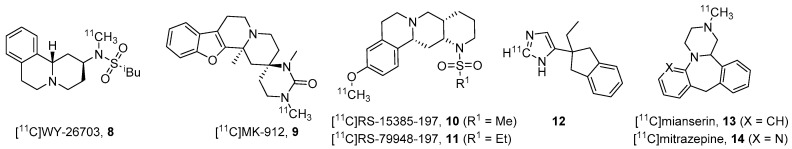
Various classes of α2-ARs antagonist radiotracers.

**Figure 3 molecules-25-04017-f003:**
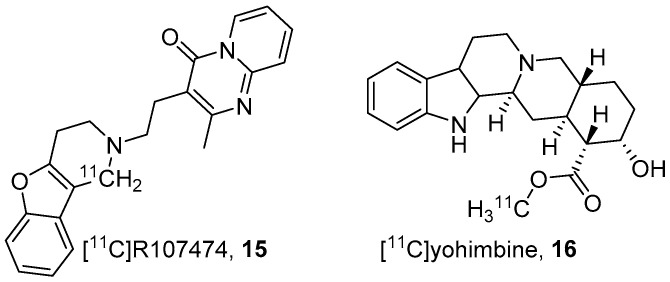
Anti-depressive & antihypertensive based α2-AR PET radiotracers.

**Figure 4 molecules-25-04017-f004:**
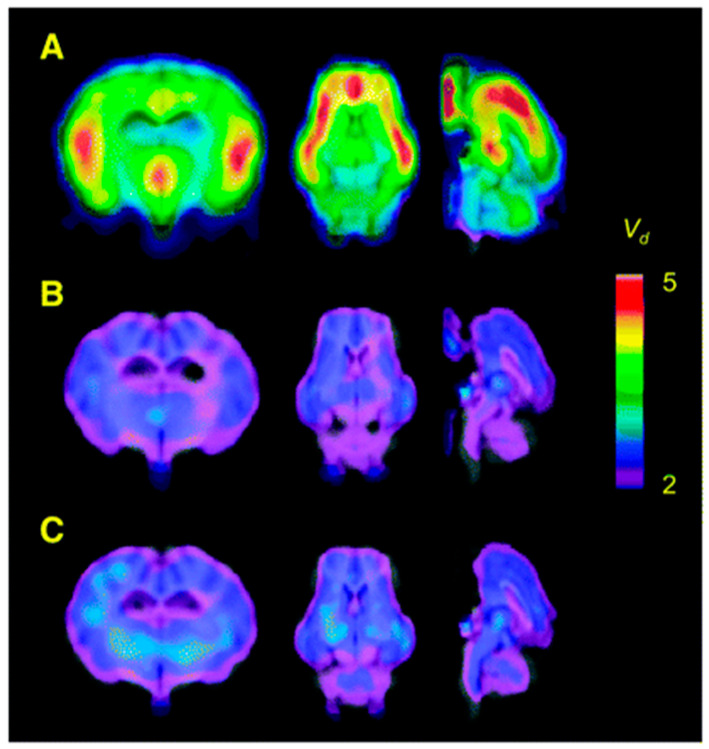
Parametric maps of **16** in living porcine brain. (**A**) Baseline study using **16** showed regional differences in its distribution. (**B**) Blocking experiment (yohimbine at 0.07 mg/kg) reduced the scale of distribution volume (*V*_d_) to ~2 mL g^−1^ in all the α2-AR bound regions. (**C**) Increased dose of yohimbine (1.6 mg/kg) had no further significant effect in comparison to the low dose (*n* = 3) Maps are superimposed on the MR image. Adapted from JNM publication by Jacobsen S, Pedersen, K.; Smith, D.F.; Jensen, S.B.; Munk, O.L.; Cumming P [97]. Permission obtained from SNMMI.

**Figure 5 molecules-25-04017-f005:**
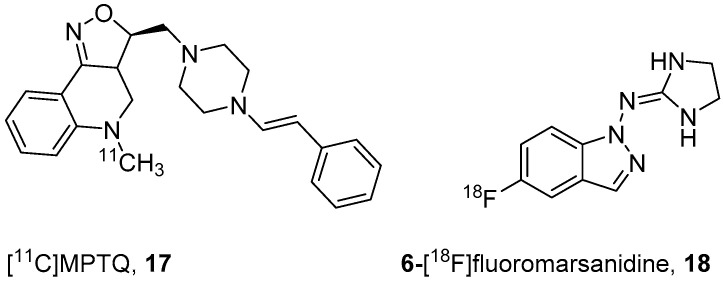
α2A-antagonist (**17**) and agonist (**18**) PET radiotracers.

**Figure 6 molecules-25-04017-f006:**
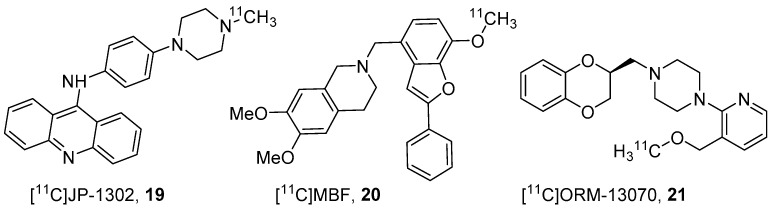
PET radiotracers for α2C-ARs.

**Figure 7 molecules-25-04017-f007:**
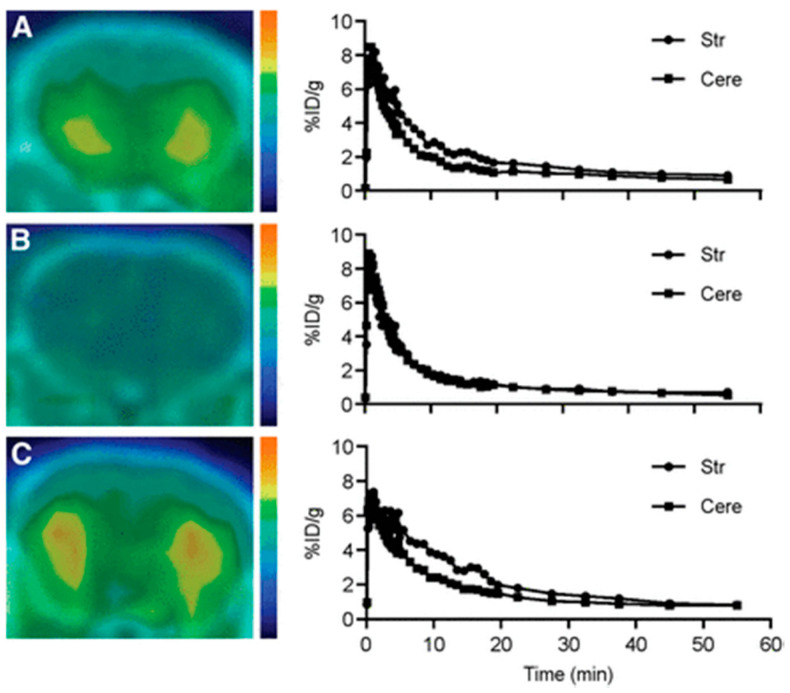
PET/CT images and time-activity curves of **21** for striatum and cerebellar cortex of (**A**) α2A KO (**B**) α2AC KO and (**C**) WT mice. Brain uptake of **21** in α2AC KO is negligible and is similar in α2A KO and WT mice with 7.8–8.1% ID/g at 1 min and 1.2% ID/g at 30 min after **21** injection. The striatum to cerebellar cortex radioactivity ratios (at 5–15 min) for α2AC KO mice did not differ and for α2A KO and WT mice are alike. Adapted from JNM publication by Arponen E.; Helin, S.; Marjamäki, P.; Grönroos, T.; Holm, P.; Löyttyniemi, E.; Någren, K.; Scheinin, M.; Haaparanta-Solin, M.; Sallinen, J.; [36]. Permission obtained from SNMMI.

**Figure 8 molecules-25-04017-f008:**
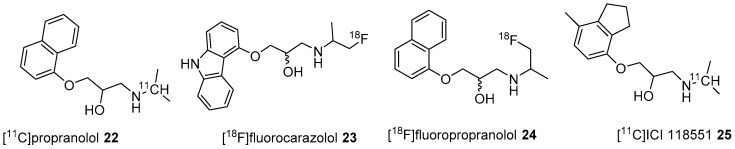
Early PET radiotracers for cerebral β-ARs.

**Figure 9 molecules-25-04017-f009:**
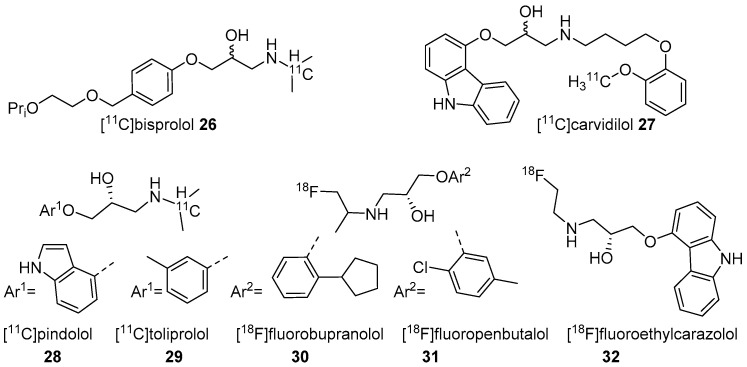
Radiotracers based on various β-AR blockers.

**Figure 10 molecules-25-04017-f010:**
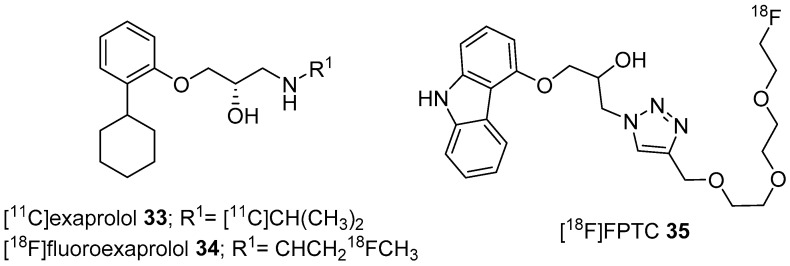
Another set of latest β-AR PET radiotracers.

**Table 2 molecules-25-04017-t002:** In vitro affinities of compounds **1** to **7** for α1-ARs, D_2_ and 5HT_2C_ receptors [73,75,76].

Compound	Receptor *K*_i_ (nM)				
	α1A	α1B	α1D	D_2_	5HT_2C_
**1**	0.37	0.33	0.66	0.45	0.55
**2**	0.23	1.1	2.0	140	330
**3**	3.0	6.0	8.6	310	1500
**4**	0.16	6.4	15	220	-
**5**	0.52	1.9	2.5	120	-
**6**	0.37	0.33	0.66	0.45	0.51
(*R*)-**7**	0.43	0.27	0.64	31	8.0
(*S*)-**7**	0.16	0.20	0.21	4.5	93

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
