# Peer review of "PET Radiotracers for CNS-Adrenergic Receptors: Developments and Perspectives"

_molecules, 2020, doi:10.3390/molecules25174017_

Round 1

Reviewer 1 Report

In this review, Alluri et al. summarized the development of PET neuroimaging probe targeting adrenergic receptors. the authors listed the current reported imaging probes as well as the introduction of PET and PET probes.

I have few minor questions/suggestions as below:

  1. For PET probe development, Bmax is important and better to know at the beginning of the development pipeline, the authors need to list if known Bmax in any brain regions;
  2. Any known expression changes of adrenergic receptors in brain diseases?
  3. line 41-34, the authors listed "....b) high 42 selectivity between subtypes or void on off-targets, c) high dynamic range in specific binding, d)
     appropriate metabolic profile,....", what are the ideal values/ranges for a good PET probe?

Reviewer 2 Report

The review paper by Alluri et al covers the development and application of radioligands for the adrenergic receptor. The review covers radioligands for the different subtypes of adrenergic receptors, and clearly briefly summarises their in vitro and in vivo properties, before describing whether the radioligands have gone on to be assessed clinically, and the outcomes of these studies. This review is well referenced and has good coverage across the literature.

I would recommend the addition of a small section which discusses the PET imaging of other components of the adrenergic system, as this is would be a useful context in which to view this review. This would not need to go into detail, but could mention the main radioligands for PET imaging of the norepinephrine transporter (NET), as well as MAO-A and MAO-B, and could provide their main research/clinical use. Without this, a reader may draw the conclusion from this review that there are no means for investigating the adrenergic system with PET. This addition would provide good context to the remainder of the review, which clearly illustrates the challenges in the development of high affinity, selective PET ligands for the adrenergic receptor subtypes.

As such, I would recommend acceptance of the review to Molecules. The following minor corrections should also be made:

  1. Line 107: A sentence on the reason why [11C]prazosin is not used as an imaging agent should be included.
  2. Throughout the manuscript, the authors refer to “specific activity” (“SA”). This should be changed to “molar activity” in line with the latest recommendations from the SRS. (See https://doi.org/10.1016/j.nucmedbio.2017.09.004). Additionally, the authors use both Ci/umol and GBq/umol in reporting this property. Either both units should be given in all cases, or one should be used throughout the manuscript.
  3. Line 118 and elsewhere: “[11C]CH3 methylation” should be “11C-methylation” (no square brackets). See the above consensus nomenclature guidelines.
  4. Throughout the manuscript, the figures containing chemical structures are not clear, with the structures of the analogues being placed very close to the core structure. This makes the figures challenging to decipher at times. The structures in the figures could be more clearly spaced.
  5. Figure 2, the legend for compound 13 should read “X = CH” and not “X = C”
  6. Reference 62 refers to an abstract – the specific abstract number should be included.
  7. Figure 3, the structure for compound 15 should read “11CH2” and not “11C”
  8. For Figure 5, compound 18 is more correctly called 6-[18F]fluoromarsanidine, as marsanidine itself does not contain a fluorine atom.
  9. Similarly, in Line 326, and lines 346-356, “[18F]carazolol” should be changed to “[18F]fluorocarazolol. This should also be corrected in Figure 9, and the associated text for compounds 30/31 which do not normally contain a fluorine atom.
  10. For Figure 7, numerical values should be included on the colour scale bar if they are available from the original reference. Additionally, in the text (line 313) it states that the TACs are from 5-15 min, however, they are from 0-60 min. Should this be that the images on the left are summed from 5-15 mins? Please clarify/correct.
  11. Line 371 “[18F]acetone” should read “[18F]fluoroacetone”
  12. Line 384: “[18F]exaprolol” should be “[18F]fluoroexaprolol”
